# Analysis of the Body Posture of Junior Kickboxers: An Observational Study

**DOI:** 10.3390/jcm13247504

**Published:** 2024-12-10

**Authors:** Krzysztof Łuniewicz-Stępniak, Aleksandra Truszczyńska-Baszak, Natalia Twarowska-Grybalow

**Affiliations:** Department of Rehabilitation, Józef Piłsudski University of Physical Education in Warsaw, 00-968 Warszawa, Poland

**Keywords:** body posture, Moiré method, physical activity, martial arts, kickboxing

## Abstract

**Background/Objectives:** The purpose of this study was to assess the body posture of kickboxing players. **Methods:** The study group included people training as kickboxers who actively participated in the training camp of the broad national junior team in the K1 and low-kick kickboxing leagues. The control group consisted of non-training people. The age of the study group ranged from 15 to 23 years. The ages of the control group participants varied from 14 to 20 years. A body posture test was performed by using the Moiré method, in accordance with the guidelines of the manufacturer of the Moiré 4G device. **Results:** The body posture test performed using the Moiré method showed statistically significant differences between the kickboxing training group and the control group. The parameters that showed statistically significant differences included the size of lordosis (KLL) and the depth of lordosis (GLL) in the lumbar spine. Higher values were observed in the kickboxing training group compared to the control group. There was no statistically significant difference between the kickboxing group and the control group in other parameters. **Conclusions:** The asymmetrical and flexed posture required during kickboxing training did not negatively influence the competitors’ spinal curvatures. Kickboxers had increased lumbar lordosis, which may be related to the specificity of the given discipline (stretching of the hamstrings and specific movements). This is a beneficial phenomenon that may reduce the risk of lower back pain. Kickboxing training did not affect the asymmetry of body posture in the sagittal plane.

## 1. Introduction

Body posture—expressed by silhouette—is an individual feature of each person. It is also an unconscious movement habit that can be corrected. It is assumed by a person in a relaxed standing position. The correct posture ensures stability, is a convenient position for movement, ensures static–dynamic performance, is ergonomic in terms of the body’s energy expenditure, does not disturb the functioning of internal organs, and meets esthetic and psychological requirements [1]. However, these criteria seem too general to be used to clearly assess and determine a patient’s condition. For this reason, during an examination, attention is paid to the general features of the correct body posture [2].

The problem of an incorrect body posture affects an increasing percentage of society [3]. During school years, body posture asymmetries affect up to 40–50% of children [4,5]. Research confirms the connection between incorrect body posture in adulthood and childhood or early adulthood [6]. Poor posture may be associated with a high incidence of back pain [7,8]. One of the key preventive measures to prevent the occurrence and worsening of posture defects in people of different ages is regular physical activity and appropriate stretching exercises [8,9,10]. A lack of appropriate prevention can contribute to pain and dysfunction in adulthood [9]. Appropriate prevention, i.e., physical activity and stretching exercises, reduces the risk of back pain persisting into adulthood [7,8]. The authors of scientific studies on this topic have observed a relationship between the size of the lumbar lordosis, the length of the hamstrings, and the frequency of back pain. Shortening of the hamstrings and a decrease in lordosis in the lumbar spine increases the frequency of back pain [10,11,12].

The authors of numerous publications attempted to assess body posture in people training in various sports [13,14,15,16]. It seems that training in martial arts may have positive benefits on the body posture [17,18,19]. Martial arts training determines the correctness of developmental processes and reduces the risk of developing many lifestyle-related diseases, such as obesity, and improves the quality of life [20,21]. However, some reports indicate that training in martial arts may increase the risk of asymmetry and posture defects [2,22,23]. Due to the forced and technically advanced movement patterns during training, changes in body posture also affect the curvatures of the spine [23].

Kickboxing is a sports discipline that has become increasingly popular within society [24]. An effective performance of individual kickboxing techniques requires a high physical fitness of the athlete. Professional kickboxers have a high level of motor skills, such as strength and power, as well as high aerobic and anaerobic capacities [25,26]. A review of the current literature does not provide an answer about the impact of kickboxing training on the body posture parameters. It is a sport that requires specific and complex skills [27]. When it is practiced, it brings with it a number of benefits in terms of aerobic and anaerobic capacity and improves motor features such as strength, speed, and agility. It also has a positive effect on muscle strength and can be considered to be an appropriate activity to improve physical fitness [18]. On the other hand, the long-term asymmetric training loads that occur in martial arts may lead to specific functional adjustments and body postures, increasing the risk of postural defects [22,23]. One of the sports that influences the size of lumbar lordosis is kickboxing. It is associated with the technique of effectively performing roundhouse and sidekicks, which create tension in the lumbar spine extensor muscles and the quadratus lumborum muscle [23]. Additionally, in people who practice kickboxing, frequent and robust kicks cause stretching of the muscles in the posterior group of the thigh, which is mentioned as a prevention of back pain [23]. Although most kickboxing athletes are male, an analysis of posture parameters in both sexes in this discipline is justified. Despite the clear anthropometric differences, the size of lumbar lordosis is referred to as a parameter that is independent of sex [28]. The Figure 1 shows the body posture of a kickboxing competitor.

The purpose of the study was to assess the body posture of kickboxers.

## 2. Materials and Methods

### 2.1. Materials

A total of 36 people were included in the study. The study group included 13 people training as kickboxers who actively participated in the training camp of the broad national junior team in the K1 and low-kick kickboxing leagues. The control group included 16 non-training people attending the John Paul II private school in Legionowo. The age of the study group ranged from 15 to 24 years. The ages of the control group participants varied from 14 to 20 years. The biometric data for the examined persons are summarized in Table 1. No statistically significant differences were observed in body weight, height, or BMI between the examined groups.

Among the participants training as kickboxers, all except one were right-handed and trained in a classic fighting position, i.e., with their right leg behind them and their left hand in front. The left-handed person trained in the reverse position. Data on training experience are summarized in the Table 2.

The study group was selected based on the following inclusion criteria: presence in the junior national team in the K1 and low-kick kickboxing leagues or active participation on their own at a team meeting and written consent from a parent or legal guardian to conduct the study. Exclusion criteria for the study group were orthopedic injury, detection of congenital defects or posture defects, chronic diseases, or disability.

The inclusion criterion for the control group was purely recreational physical activity and age in the range 14–20 years. Exclusion criteria were orthopedic injury preventing kickboxing training within the last year, detection of congenital defects or posture defects, chronic diseases, or disability. The control group was selected to match the age, gender, height, and body weight of the study group.

### 2.2. Methodology

The study was performed in March 2022 using the observational research method. Before starting the body posture examination, consent to the examination and an original questionnaire were completed. The questionnaire contained questions about training experience, number of training units per week, dominant hand, and past injuries. The body posture assessment was carried out by using the Moiré method with help of the Moiré 4th Generation system (a specialist piece of equipment for the computer assessment of body posture—CQ Elektronic System in Poland). The test and results were prepared in accordance with the device manufacturer’s guidelines. The study was conducted by one researcher, which limited the occurrence of technical errors. The research methodology was consistent with the assumptions of the Declaration of Helsinki. The research began after obtaining consent from the Senate Research Committee on Research Ethics. Participation in the study was voluntary. The subjects were informed about the purpose and course of the study before participating in it. Each participant was informed about the possibility of withdrawing at any stage of the study.

### 2.3. The Moiré Method

The Moiré method is commonly used in body posture testing due to its availability, non-invasiveness, and low cost [29]. The reliability and validity of the Moiré method have been confirmed in many studies over recent years [30,31,32].

The body posture examination was carried out based on the analysis of selected parameters that characterize the following:Alpha angle—the inclination of the lumbosacral spine;Beta angle—the inclination of the thoracolumbar section;Gamma angle—the inclination of the upper thoracic spine;KKP—the size of kyphosis in the thoracic spine;KLL—the size of lordosis in the lumbar spine;KPT—the torso inclination angle;GLL—the depth of lordosis in the lumbar spine;UL—the difference in height between the right and left shoulder blades;UB—the difference in depth between the right and left shoulder blades;OL—the difference in the distance between the right and left shoulder blades in relation to the spine line;KLB—the angle of inclination of the right and left shoulder;TT—the difference in height of the right and left waist triangle;TS—the difference in width between the right and left waist triangles;KNM—the angle of inclination of the posterior iliac spines in the pelvis;KSM—the angle of twisting of the posterior iliac spines in the pelvis.

Figure 2 shows sample photo of an examined person with anatomic points for calculating postural parameters marked

### 2.4. Statistical Analysis

Statistical calculations were performed in the Statistica version 13 program from StatSoft in Poland. The Shapiro–Wilk test was used to assess the distribution. For biometric data, Hotelling’s T-squared distribution was performed for independent variables. The U Mann–Whitney test was used to assess the parameters determining body posture. The magnitude of differences was expressed by Cohen’s *d* effect sizes. The level of statistical significance was set at *p* ≤ 0.05.

## 3. Results

The body posture examination carried out by using the Moiré method allowed for an assessment of the angles determining the size of spinal curvature in the participants. There was no statistically significant difference between the kickboxing group and the control group in the Alpha, Beta, and Gamma parameters. There was no statistically significant difference in the size of kyphosis in the thoracic spine (KKP) and the torso inclination angle (KPT) between the study groups.

The parameters that showed statistically significant differences included the size of lordosis (KLL) and the depth of lordosis (GLL) in the lumbar spine. Higher values were observed in the kickboxing training group compared to the control group.

The parameters describing the shape of the spine curvatures are summarized in Table 3.

Data on parameters describing differences in the angles of the lower blades were also collected. The analysis of the above parameters did not show a statistically significant difference between the kickboxing group and the control group in the UL parameter measured in millimeters or degrees. There was no difference in the UB parameter in millimeters or degrees and no difference in the OL parameter.

There was also no statistically significant difference in the shoulder position (KLB) in millimeters or degrees. Also, the difference in TT and TS was not statistically significant between the groups.

An analysis of the pelvic position in both groups was also performed. There was no statistically significant difference between the groups in the KSM parameter in millimeters (*p* = 0.552) or degrees (*p* = 0.616) or in the pelvic inclination angle (KNM) in millimeters (*p* = 0.232) or degrees (*p* = 0.315).

Data regarding parameters describing the position of the shoulders, shoulder blades, waist triangle, and pelvis are summarized in Table 4.

## 4. Discussion

The analysis of the current knowledge on this topic did not clearly indicate that a relationship between training martial arts and body posture exists. There is also a lack of research on the impact of kickboxing training on the body posture. Some studies indicate a positive impact of martial arts training on body posture [18], but there are also studies that contradict this [22,23]. There has also been scientific evidence confirming the problem of postural defects in adolescents regardless of the level of physical activity [33,34]. Additionally, an important fact is that testing the same group training a selected sport twice is the only way to actually confirm the impact of training on body posture parameters. Such a study was performed by Grabara, which confirmed the negative impact of training on parameters that determine the position of the shoulder blades and pelvis [14].

Based on our own research, a statistically significant difference was observed in the following parameters: KLL (the size of lordosis in the lumbar spine) and GLL (the depth of lordosis in the lumbar spine). Higher values in both of the above parameters in the kickboxing group indicate increased lordosis in the lumbar spine compared to the control group.

When analyzing linear parameters using the Moiré method, it should be assumed that a difference between individual bone points in the range of 5–10 mm indicates moderate asymmetry while a difference greater than 10 mm indicates significant asymmetry. For parameters calculated using angular values, the range between 1.5 and 30 means moderate asymmetry while and values above 30 mean significant asymmetry [30]. Based on our own research, the mean values of the parameter determining the angle of pelvic inclination indicate that there was no significant asymmetry in either group. Higher angle values indicate greater pelvic asymmetry and such results were observed in the control group.

Based on the authors’ previous research, the size of the angles responsible for the size of spine curvature was assessed and the angle of the torso inclination was analyzed. People who trained in kickboxing, compared to people who did not train in it, had deeper lumbar lordosis and a tilted pelvis with the left anterior superior iliac spine being higher. The average UL values (the difference in the height of the lower angles of the shoulder blades) showed a different elevation of the shoulder blades in both groups, but these values were not statistically significant. From the above, it can be concluded that the characteristics of the body posture of the people training in kickboxing did not differ statistically significantly from those of the control group. The increase in lumbar lordosis in the study group is related to the specificity of individual movement patterns during training. Effective kicks require appropriate elasticity of the hamstrings. The authors of numerous studies have proven that a deepening of lumbar lordosis and the lack of a shortening of the hamstrings in the thigh have an impact on the lack of back pain. It can, therefore, be stated that practicing kickboxing is a preventive measure for back pain [10,11,12].

In his work, Mrozkowiak [2] examined the body posture of 424 girls and 355 boys aged 15 by using the Moiré method. The aim of the study was to determine, among others, the normative values that define the shape of the spine. Comparing the results of our own research to those of Mrozkowiak, it can be concluded that the values describing the size of lordosis in the lumbar section—despite statistically significant differences—are within the normal range in both the kickboxing training group and in the control group. The differences are therefore not clinically significant.

The impact of training on posture was confirmed by Dobrescu’s study [22], which analyzed judo, mixed martial arts, and wrestlers. Those who trained in these sports, despite having a less leaning forward posture, had reduced lumbar lordosis and increased thoracic kyphosis. They were also characterized by a raised left scapula and left anterior superior iliac spine. In kickboxing, the dominant upper limb is usually farther from the opponent. According to a previously conducted survey, all the participants, except for one person from the study group, fought in a position with their left upper limb closer to their opponent. Our study showed an elevation of the left scapula in the training group and an elevation of the right scapula in the non-training group, but these relationships were not statistically significant. It can be hypothesized that the shoulder blade of the upper limb, which was closer to the opponent, would be raised higher than the opposite one. This hypothesis was not confirmed in the study by Mrozkowiak et al. [2]. In that study, the posture of the wrestlers was characterized by a much higher raised left shoulder, where the dominant upper limb was closer to the opponent in the fighting position. In turn, in the study by Socha et al. [35], it was observed that the techniques used in martial arts had a greater impact on the anthropometric parameters of female judo athletes than the use of the dominant hand. This could have occurred due to the specificity of this discipline.

Mrozkowiak et al. [2] also analyzed body posture in terms of the curvatures and angle of inclination in the spine. The posture of the judo players in the sagittal plane showed a slight deepening of the lumbar and thoracic lordosis. Wrestlers in the sagittal plane, in relation to the posture of judo competitors and volleyball players, had a greater angle of lumbar lordosis and a smaller inclination of the lumbosacral section, a smaller angle of thoracic kyphosis, and a smaller depth of lumbar lordosis. Comparing the results of our own study to the characteristics of the judo and wrestling competitors in the study by Mrozkowiak et al., the kickboxing competitors showed a greater depth of lumbar lordosis.

In the work of Domaradzki et al. [23], it was found that the long-term, asymmetric training load that occurs when practicing martial arts could lead to specific functional adjustments and body posture adaptations. Among people training in kickboxing, it was observed that the angle of inclination of the thoracolumbar and lumbosacral sections increased significantly as a result of training compared to other groups of athletes, which could have been a factor increasing the risk of posture defects. This was not confirmed by the results of our own study. A greater angle of thoracolumbar inclination and a greater torso inclination were observed in the control group.

The issue of the impact of martial arts on the body posture is rarely discussed in the current literature. The value of our own research is that it includes a less popular sport such as kickboxing. An additional advantage of the study is that the body posture test by using the Moiré method was performed and developed by only one researcher. This excluded technical and measurement errors resulting from differences in the way bone points were marked on the subjects’ bodies and different interpretations of the results obtained during the data processing.

The kickboxing group was characterized by having the correct body posture. The directions for future research should be to increase the number of subjects practicing kickboxing and repeat the study after a few years. Only these measurements can clearly indicate the impact of kickboxing training on body posture

The main limitation of the study is that the people practicing kickboxing were examined only once. Only double examinations would allow us to clearly determine the impact of training in a selected sport on body posture. Another limitation of the study is that the study group was too uneven in terms of the age range and length of training experience of the subjects. Although the authors of the study tried to select the most homogeneous control group in terms of anthropometric parameters, there is a discrepancy in age between the above groups. Despite the lack of difference in the clinical context, it is worth considering this problem when designing the next study. The direction of future research should be to expand the research group by including more people of a similar age, with a similar length of training experience, and to examine the group twice after a specified period of time.

## 5. Conclusions

The asymmetrical and flexed posture required during kickboxing training did not negatively influence the spinal curvatures of the kickboxing competitors.The kickboxers had increased lumbar lordosis, which may be related to the specificity of the given discipline (stretching of the hamstrings and specific movements). This is a beneficial phenomenon that may reduce the risk of lower back pain.Kickboxing training did not affect the asymmetry of body posture in the sagittal plane.

## Figures and Tables

**Figure 1 jcm-13-07504-f001:**
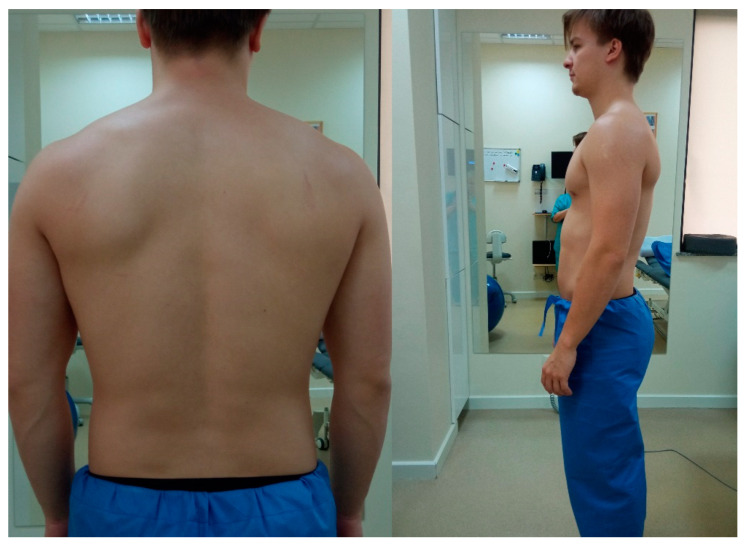
A photo of the body posture of one of the authors (KŁS) of the article, who is a kickboxing competitor.

**Figure 2 jcm-13-07504-f002:**
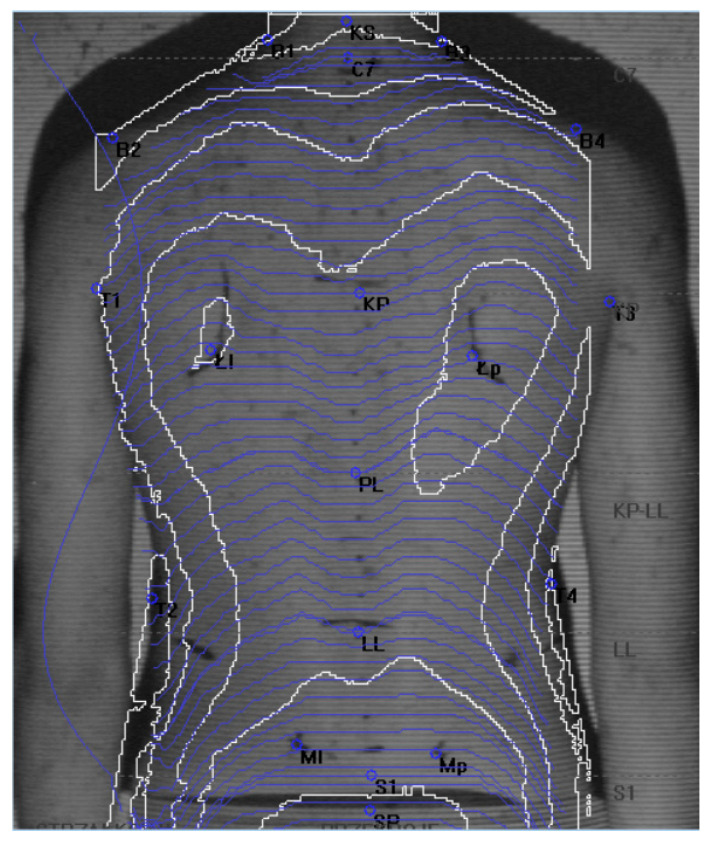
Sample photo of an examined person with anatomic points for calculating postural parameters marked. Source: own material.

**Table 1 jcm-13-07504-t001:** Biometric data of the surveyed people.

	Group		*p*-Value	Cohen’s *d*
Variables	Kickboxing	Control		
Age [years]	18.33 ± 2.85	16.77 ± 1.41	0.042	0.69
Body height [cm]	178.18 ± 7.40	177.21 ± 7.60	0.773	0.13
Body weight [kg]	73.41 ± 9.94	68.21 ± 9.94	0.267	0.52
BMI [kg·m^2^]	23.00 ± 3.48	21.68 ± 3.67	0.275	0.37

**Table 2 jcm-13-07504-t002:** Training experience data.

	Group
Variables	Kickboxing
Length of athletic career [years]	2.09 ± 1.04
Length of training [years]	6.6 ± 2.66
Number of trainings per week [units]	4.16 ± 0.99

**Table 3 jcm-13-07504-t003:** Characteristics of the parameters describing the shape of the spine curvatures.

Variables [°]	Group		Parameter			
	Kickboxing [n = 17]	Control [n = 19]	*p*-Value	Z	U	Cohen’s *d*
Alpha	11.29 ± 4.98	9.35 ± 4.32	0.315	−1.00	129.5	0.42
Beta	7.79 ± 3.62	5.42 ± 3.59	0.076	−1.76	105.5	0.66
Gamma	14.34 ± 3.98	15.93 ± 7.54	0.876	0.14	156.5	0.26
KKP	157.94 ± 7.07	160.70 ± 9.32	0.129	1.51	113.5	0.33
KPT	5.21 ± 1.74	7.27 ± 5.09	0.066	1.82	103.5	0.54
KLL	167.27 ± 7.18	160.99 ± 7.38	0.042	2.01	97.50	0.86
GLL	13.41 ± 8.55	8.21 ± 4.48	0.033	−2.11	94.50	0.76

Note: mean/SD.

**Table 4 jcm-13-07504-t004:** Parameters describing the position of the shoulders, shoulder blades, waist triangle, and pelvis.

Variables	Group		Parameter			
	Kickboxing [n = 17]	Control [n = 19]	*p*-Value	Z	U	Cohen’s *d*
UL [mm]	7.06 ± 5.54	7.45 ± 9.14	0.594	−0.53	144.5	0.05
UL [°]	2.82 ± 2.43	3.34 ± 4.45	0.616	−0.51	145.0	0.15
UB [mm]	6.16 ± 7.76	5.29 ± 4.57	0.707	0.37	149.5	0.14
UB [°]	1.78 ± 1.81	2.10 ± 1.80	0.490	0.68	139.5	0.18
OL [mm]	13.31 ± 17.24	6.55 ± 6.14	0.129	−1.52	113.0	0.52
KLB [mm]	5.23 ± 5.42	7.41± 6.82	0.346	0.96	131.0	0.35
KLB [°]	0.86 ± 0.91	1.43 ± 1.29	0.175	1.36	118.5	0.51
TT [mm]	10.27 ± 8.07	12.29 ± 10.23	0.531	0.62	141.5	0.22
TS [mm]	7.90 ± 6.07	9.23 ± 6.07	0.531	0.63	141.0	0.22
KSM [mm]	3.07 ± 3.12	3.15 ± 2.05	0.552	0.59	142.5	0.03
KSM [°]	2.21 ± 2.09	2.18 ± 1.41	0.616	0.49	142.5	0.02
KNM [mm]	2.34 ± 2.48	3.08 ± 2.64	0.232	1.21	123.5	0.29
KSM [°]	1.66 ± 1.64	2.16 ± 1.78	0.315	1.01	129.5	0.29

Note: mean/SD.

## Data Availability

The raw data supporting the conclusions of this article will be made available by the authors on request.

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
