# Peer review of "Analysis of the Body Posture of Junior Kickboxers: An Observational Study"

_jcm, 2024, doi:10.3390/jcm13247504_

Round 1

Reviewer 1 Report

Comments and Suggestions for Authors

Abstract

Page 1, Line 14-16: How different in lumbar lordosis? More specific.

Page 1, Line 18-21: The conclusion seems to contradict the results mentioned in the abstract.

Introduction

Page 1, Line 45-56: A lot of sports involves the asymmetric body movement depending on the hand dominancy, such as throwing and shooting a ball, swinging a racket, fighting with a sword and so on. The research background which made the authors focusing on kickboxing is poorly described. Any medical problems related to asymmetric body movement in kickboxing? It should be clearly mentioned.

Methods

Page 2, Line 72-75: The exclusion criteria for the kickboxing group should be provided also. Were the athletes with any musculoskeletal symptoms, which possibly affect body posture, excluded?

Page 2, Line 79-80: There is an age mismatch between 2 groups, with a statistical significance. Why don’t you achieve age matching?

Page 3, Line 102-119: Since most of readers are not familiar with Moire method, the reviewer recommends adding some figures with Moire images which indicate how to measure the parameters. They help the readers understanding the significance.

Results

Well written.

Page 134-135: What does it mean that the size and depth of lumbar lordosis increased in the kickboxing group. Please explain that using anatomical and orthopaedic terms for the readers who are unversed in Moire.

Discussion

Logical discuss is required to understand possible causes of and factors related to the significantly greater and deeper lordosis of lumbar spine in the kickboxing group, compared to the control group.

Conclusion

Page 6, Line 249-245: There was a significant difference in lumbar lordosis between the kickboxing group and the control group. The authors cannot discard the important fact obtained from this study.

Author Response

Thank you very much for your thorough substantive and editorial evaluation. All comments and tips are extremely important and will serve to improve the quality of the article, for which we are grateful. Below is a list of changes that we have included in the publication.

  • Revision

  1. Abstract: Page 1, Line 14-16: How different in lumbar lordosis? More specific. Page 1, Line 18-21: The conclusion seems to contradict the results mentioned in the abstract.

Added information about the results: „The parameters that showed statistically significant differences included: the size of lordosis (KLL) and the depth of lordosis (GLL) in the lumbar spine. Higher values were observed in the kickboxing training group compared to the control group.”

  1. Abstract: Page 1, Line 18-21: The conclusion seems to contradict the results mentioned in the abstract.

Conclusion reworded: „Kickboxers had increased lumbar lordosis, which may be related to the specificity of the given discipline (stretching of the harmstrings and specific movements). This is a beneficial phenomenon that may reduce the risk of lower back pain.”

  1. Introduction: Page 1, Line 45-56: A lot of sports involves the asymmetric body movement depending on the hand dominancy, such as throwing and shooting a ball, swinging a racket, fighting with a sword and so on. The research background which made the authors focusing on kickboxing is poorly described. Any medical problems related to asymmetric body movement in kickboxing? It should be clearly mentioned.

We agree that the introduction contained not enough information about the specifics of kickboxers' training and body posture and that it lacked information about the importance of lumbar lordosis in the spine. The introduction has been expanded with following sentences:

„One of the sports that influences the size of the lumbar lordosis is kickboxing. It is associated with the technique of effectively performing roundhouse and sidekicks, which forces tension in the lumbar spine extensor muscles and the quadratus lumborum muscle [23]. Additionally, in people who practice kickboxing, frequent and robust kicks cause stretching of the muscles of the posterior group of the thigh, which is mentioned as a prevention of back pain [23]. Although most kickboxing athletes are male, the analysis of parameters in both sexes in this discipline is justified. Despite the obvious anthropometric differences, the size of the lumbar lordosis is mentioned as a parameter independent of sex [28].”

  1. Material and Methods: Page 2, Line 72-75: The exclusion criteria for the kickboxing group should be provided also. Were the athletes with any musculoskeletal symptoms, which possibly affect body posture, excluded?

Added information about the exclusion criteria for the study: “Exclusion criteria for the study group are: orthopedic injury, detection of congenital defects or posture defects, chronic diseases or disability.”

  1. Material and Methods: Page 2, Line 79-80: There is an age mismatch between 2 groups, with a statistical significance. Why don’t you achieve age matching?

The difference between the age of the respondents is explained in the discussion chapter in the section Limitations of the study.

  1. Material and Methods: Page 3, Line 102-119: Since most of readers are not familiar with Moire method, the reviewer recommends adding some figures with Moire images which indicate how to measure the parameters. They help the readers understanding the significance.

A figure showing the examined person has been added, which will help the reader understand the method presented in the study.

  1. Results

Thank You very much for your opinion. In response to the remark regarding the lack of explanation of the effect of the size and depth of lumbar lordosis on the kickboxing group, the following sentence has been added in the discussion chapter: „Higher values ​​in both of the above parameters in the kickboxing group indicate increased lordosis in the lumbar spine compared to the control group.”

  1. Discussion: Logical discuss is required to understand possible causes of and factors related to the significantly greater and deeper lordosis of lumbar spine in the kickboxing group, compared to the control group.

Added the following excerpts:

„The increase in lumbar lordosis in the study group is related to the specificity of individual movement patterns during training. Effective kicks require appropriate elasticity  of the harmstrings. The authors of numerous studies have proven that the deepening of lumbar lordosis and the lack of shortening of the harmstrings of the thigh have an impact on the lack of back pain. It can therefore be stated that practicing kickboxing is a preventive measure for back pain [10-12].”

  1. Conclusion: Page 6, Line 249-245: There was a significant difference in lumbar lordosis between the kickboxing group and the control group. The authors cannot discard the important fact obtained from this study.

The following conclusion has been added: „Kickboxers had increased lumbar lordosis, which may be related to the specificity of the given discipline (stretching of the harmstrings and specific movements). This is a beneficial phenomenon that may reduce the risk of lower back pain.”

Reviewer 2 Report

Comments and Suggestions for Authors

Could you add the study type to the title for clarity? For example, including "Observational Study" or "Cross-Sectional Analysis" would help readers immediately understand the nature of the research.

Narrative Structure

Lines 27-36: It would strengthen the introduction to discuss the variability of risk based on sport type and intensity, as this may help contextualize kickboxing within the broader spectrum of martial arts.

Lines 103-132: Consider adding real-life case studies or anecdotal data related to kickboxing athletes or similar sports to better illustrate points on posture and injury risks.

Lines 181-236: An expanded focus on preventative measures for posture-related issues, especially conditions like thoracic outlet syndrome (TOS), would enhance the practical value of your findings.

Lines 33-36: When discussing the significance of finding no major postural differences, provide additional context on how these findings align with or challenge existing research on postural asymmetry and corrective measures.

Lines 74-102: Since you focus on junior athletes, addressing potential postural differences or specific risk factors in both male and female athletes could create a more balanced overview.

Consider smoothing transitions between sections to improve the manuscript’s readability.

Ensure consistent formatting and use of subheadings for clear distinctions between sections, which would enhance the organizational flow.

Description of Methods

Sufficiency of Method Explanations:

Lines 407-437: Providing additional details on the measurement techniques for body posture and specific parameters (e.g., Moiré method setup and analysis) would strengthen this section. Clarifying how each parameter relates to the study’s objectives could also improve comprehension.

Recommendations for Improvements:

Figure 1: Enhance this figure by creating a more detailed flowchart of factors influencing postural asymmetry and injury risk in kickboxing.

Table 1: Including data such as kickboxing sport type, length of training or athletic career, and any injury-related incidents would provide more depth.

Figure 2: Use vivid colors and clearer labels to make the figure more readable, especially for visually oriented readers.

Author Response

Thank you very much for your thorough substantive and editorial evaluation. All comments and tips are extremely important and will serve to improve the quality of the article, for which we are grateful. Below is a list of changes that we have included in the publication.

2 Revision

  1. The title has been changed to: “Analysis of the Body Posture of Junior Kickboxing Players. Observational Study.”
  2. Lines 27-36: It would strengthen the introduction to discuss the variability of risk based on sport type and intensity, as this may help contextualize kickboxing within the broader spectrum of martial arts. Lines 103-132: Consider adding real-life case studies or anecdotal data related to kickboxing athletes or similar sports to better illustrate points on posture and injury risks.

The introduction and discussion chapter have been extended with information on the impact of kickboxing training on body posture (especially on the curvature of the spine) and on the importance of lordosis in the lumbar spine. Chapters have been edited using 6 new citations, which has helped to expand the knowledge gap.

  1. Lines 181-236: An expanded focus on preventative measures for posture-related issues, especially conditions like thoracic outlet syndrome (TOS), would enhance the practical value of your findings.

The work has been expanded to include knowledge about the structure and importance of the lumbar spine. A review of the literature has shown that changes in body posture in kickboxers relate to this parameter.

  1. Lines 33-36: When discussing the significance of finding no major postural differences, provide additional context on how these findings align with or challenge existing research on postural asymmetry and corrective measures.

Added information as suggested above: ” The increase in lumbar lordosis in the study group is related to the specificity of individual movement patterns during training. Effective kicks require appropriate elasticity  of the harmstrings. The authors of numerous studies have proven that the deepening of lumbar lordosis and the lack of shortening of the harmstrings of the thigh have an impact on the lack of back pain. It can therefore be stated that practicing kickboxing is a preventive measure for back pain [10-12].”

  1. Lines 74-102: Since you focus on junior athletes, addressing potential postural differences or specific risk factors in both male and female athletes could create a more balanced overview.
  2. The information was edited based on the technique of performing movements in kickboxing. In athletes under the influence of training, lordosis in the lumbar region may increase, but clinically this is a positive phenomenon that can be considered a prevention of back pain. Information has been added based on new citations
  3. The following sentences have been added: „Although most kickboxing athletes are male, the analysis of parameters in both sexes in this discipline is justified. Despite the obvious anthropometric differences, the size of the lumbar lordosis is mentioned as a parameter independent of sex [28].”
  4. Consider smoothing transitions between sections to improve the manuscript’s readability. Ensure consistent formatting and use of subheadings for clear distinctions between sections, which would enhance the organizational flow.

Individual chapters have been edited and subchapters have been added to the chapter Material and Methods

  1. Description of Methods
  2. Sufficiency of Method Explanations:
  3. Lines 407-437: Providing additional details on the measurement techniques for body posture and specific parameters (e.g., Moiré method setup and analysis) would strengthen this section. Clarifying how each parameter relates to the study’s objectives could also improve comprehension.

A figure showing the examined person has been added, which will help the reader understand the method presented in the study

  1. Recommendations for Improvements:

A figure on the Moiré method has been added and colors in tables have been added. In order to improve the clarity of the data, a separate table has been added - Table 2, containing data on training experience.

Round 2

Reviewer 1 Report

Comments and Suggestions for Authors

The responses for the first review queries are appropriate and the manuscript has been improved.

In some sentences, mistyped "harmstrings" should be corrected to "hamstrings". 

Author Response

Thank you for your positive review. The wrong word has been corrected throughout the work.